# Propionate and Butyrate Inhibit Biofilm Formation of *Salmonella* Typhimurium Grown in Laboratory Media and Food Models

**DOI:** 10.3390/foods11213493

**Published:** 2022-11-03

**Authors:** Jiaxiu Liu, Wenxiu Zhu, Ningbo Qin, Xiaomeng Ren, Xiaodong Xia

**Affiliations:** 1National Engineering Research Center of Seafood, School of Food Science and Technology, Dalian Polytechnic University, Dalian 116034, China; 2College of Food Science and Engineering, Northwest A&F University, Yangling 712100, China

**Keywords:** propionate, butyrate, biofilm, antimicrobial activity, *Salmonella* Typhimurium

## Abstract

*Salmonella* is among the most frequently isolated foodborne pathogens, and biofilm formed by *Salmonella* poses a potential threat to food safety. Short-chain fatty acids (SCFAs), especially propionate and butyrate, have been demonstrated to exhibit a beneficial effect on promoting intestinal health and regulating the host immune system, but their anti-biofilm property has not been well studied. This study aims to investigate the effects of propionate or butyrate on the biofilm formation and certain virulence traits of *Salmonella*. We investigated the effect of propionate or butyrate on the biofilm formation of *Salmonella enterica* serovar Typhimurium (*S.* Typhimurium) SL1344 grown in LB broth or food models (milk or chicken juice) by crystal violet staining methods. Biofilm formation was significantly reduced in LB broth and food models and the reduction was visualized using a scanning electron microscope (SEM). Biofilm metabolic activity was attenuated in the presence of propionate or butyrate. Meanwhile, both SCFAs decreased AI-2 quorum sensing based on reporter strain assay. Butyrate, not propionate, could effectively reduce bacterial motility. Bacterial adhesion to and invasion of Caco-2 cells were also significantly inhibited in the presence of both SCFAs. Finally, two SCFAs downregulated virulence genes related to biofilm formation and invasion through real-time polymerase chain reaction (RT-PCR). These findings demonstrate the potential application of SCFAs in the mitigation of *Salmonella* biofilm in food systems, but future research mimicking food environments encountered during the food chain is necessitated.

## 1. Introduction

*Salmonella* is one of the most common foodborne pathogens, which is responsible for 94 million cases of gastroenteritis every year and led to 155,000 deaths around the world in 2016 [1,2]. *Salmonella* is frequently divided into zoonotic non-typhoidal *Salmonella* serovars and human-adapted typhoidal *Salmonella* serovars. Non-typhoidal *Salmonella* mainly causes gastrointestinal disorders, while typhoidal *Salmonella* serovars could cause severe extraintestinal diseases [3].

A biofilm is a community of microorganisms and extracellular polymeric substances formed by microorganisms to increase their resistance to the extreme environment [4]. Bacterial pathogens could adhere to and grow biofilm on biological or non-biological substances [5]. *Salmonella* is a common biofilm-forming pathogen, and the survival of *Salmonella* relies partly on the formed biofilm, which makes bacteria more resistant to antimicrobial agents than planktonic bacteria [6,7]. It has been reported that the biofilm lifestyle of *Salmonella* is more favorable for the growth of *Salmonella* in the host than in planktonic cells [8]. In the food industry, biofilm is formed by *Salmonella* on different surfaces, such as food processing equipment, food packaging materials, and even cooking utensils [9]. Those biofilms may cause product contamination and human infection, as well as decreased fluid flow during processing [10]. In addition to thermal treatment and chemical disinfectants, researchers have been searching for alternative strategies using natural antimicrobials to combat biofilm in the food industry.

Quorum sensing (QS) is an important system in *Salmonella*, and QS molecules, such as autoinducer-2 (AI-2), regulate a variety of cellular processes, including motility, biofilm formation, and virulence [11]. Furthermore, mobility is recognized as an important factor for biofilm formation [12]. Pathogens could use their motility to attach to the surfaces of living or non-living things, which contributes to the first stage of biofilm formation [13]. Since *Salmonella* is commonly detected in various foods (such as chicken) and causes a large number of foodborne diseases, it poses a great challenge for the food industry [5,8]

Short-chain fatty acids (SCFAs) produced by gut microbiota fermentation of dietary polysaccharides show substantial effects on host immunity and can mediate colonization resistance against bacterial enteric infection [14,15]. Although the beneficial effects of SCFAs on human health have been extensively explored in vivo, their antibacterial impacts on foodborne pathogens have been rarely examined [4]. Recently, SCFAs have been tested for their effect on *Salmonella* biofilm, but the experiment was carried out using laboratory media, rather than real food media [16,17,18].

To date, the laboratory medium has been widely used to investigate the inhibitory effect of the substance on biofilm production [19]. Some kinds of bacteria need a reasonable food matrix in which to grow better, and laboratory mediums cannot mimic the food conditions, making laboratory-oriented results inapplicable in the food industry. It is important to explore the accurate effect of biofilm formation in the food matrix [20,21].

In this study, we mainly investigated the inhibitory effect of two major SCFAs (propionate and butyrate) on the biofilm formation of *Salmonella enterica* serovar Typhimurium (*S.* Typhimurium) SL1344 grown in laboratory medium (LB broth) and two food models (milk and chicken broth). Meanwhile, the effect of two SCFAs on bacterial motility, AI-2 quorum sensing, and expression of genes related to biofilm formation and invasion were also explored.

## 2. Materials and Methods

### 2.1. Reagents

Propionate (CAS:137-40-6) and butyrate (CAS:156-54-7) were purchased from Aladdin (Shanghai, China). All other chemicals were of analytical grade.

### 2.2. Bacterial Strains and Culture Conditions

*S.* Typhimurium SL1344 and *Vibrio harveyi* BB170 were purchased from the American Type Culture Collection (ATCC) and stored in our laboratory. *S.* Typhimurium SL1344 and *Vibrio harveyi* BB170 were transferred into LB broth, cultured at 37 °C overnight at 120 rpm, then the overnight culture was diluted to an optical density (OD) at 600 nm to 0.5 (approximately 10^8^ CFU/mL) by a microplate reader (Tecan, Männedorf, Switzerland) for further use.

### 2.3. Preparation of Milk and Chicken Broth for Biofilm Growth

Fresh milk was purchased from the local supermarket. For eliminating the possible pathogens, milk was pasteurized at 75 °C for 30 min, then stored at 4 °C before use.

The sterile chicken broth was prepared as described previously [22]. Briefly, the fresh chicken breast was purchased from the local supermarket (Tesco, Dalian, China), and transported to the laboratory on ice. Then, 500 g of chicken breast was processed into small pieces and homogenized with 500 mL 0.1 M phosphate buffer solution (PBS) to get chicken juice. Then, the juice was collected and boiled at 100 °C for 20 min. Afterward, the collected juice was divided into sterile bottles and stored at 4 °C before use.

### 2.4. Minimum Inhibitory Concentrations (MICs) and Sub-Inhibitory Concentration (SICs) Assay

The MICs of propionate or butyrate against the *Salmonella* strains were determined with the micro-broth dilution method based on the Clinical and Laboratory Standards Institute guidelines [23]. *S.* Typhimurium SL1344 was prepared as described in Section 2.2. Bacteria were incubated with LB broth supplemented with various concentrations of propionate or butyrate (from 0 mg/mL to 128 mg/mL) at a final concentration of 5 × 10^5^ CFU/mL in a 96-well plate. The plate was incubated at 37 °C for 24 h. The MIC is defined as the lowest concentration of antimicrobial agent that inhibits the visible growth of a microorganism and is determined by the naked eye after 24 h incubation.

The SICs of propionate or butyrate against the *Salmonella* strains were determined by Bioscreen C Automated Microbiology Growth Curve Analysis System (Labsystems, Helsinki, Finland). *S.* Typhimurium SL1344 was prepared as described in Section 2.2. Bacteria were incubated with LB broth supplemented with various concentrations of propionate or butyrate (from 0 mg/mL to 64 mg/mL) at a final concentration of 5 × 10^5^ CFU/mL in a 100-well plate. The plate was incubated at 37 °C for 48 h and was monitored automatically every 1 h at OD600 nm. The sub-inhibitory concentrations (SICs), which did not affect the growth of microorganisms, were chosen based on the growth curves of *S.* Typhimurium SL1344 treated with different concentrations of SCFAs.

### 2.5. Specific Biofilm Formation Inhibition Assay

Biofilm formation was examined by the crystal violet (CV) staining method as previously described with some modifications [5]. The overnight *S.* Typhimurium SL1344 was centrifuged at 8000× *g* for 5 min, then re-suspended in fresh LB broth or food broths (milk and chicken broth), respectively. *S.* Typhimurium SL1344 (OD600 nm = 0.5) was incubated into 96-well plates supplemented with different concentrations of propionate or butyrate (0.5 mg/mL, 1.0 mg/mL, and 2.0 mg/mL) for 24 h or 48 h. The untreated *S.* Typhimurium SL1344 was used as a control. At each time point, bacterial growth was determined by measuring OD 630. The growth media was carefully removed, and each well was rinsed twice with sterile distilled water to remove unattached bacteria. Next, biofilms were stained with 200 μL of 0.1% (*v*/*v*) CV solution for 20 min. The CV solution was discarded, and each well was rinsed twice with sterile distilled water to remove unbound colorant. The plate was air-dried for 40 min, the stained biofilm was solubilized in 200 μL of 33% (*v*/*v*) glacial acetic acid for 20 min and measured at OD570 nm. Specific biofilm formation (SBF) was calculated by attaching and staining bacteria (OD570 nm) normalized with cell growth (OD630 nm) by a microplate reader.

### 2.6. Scanning Electron Microscopy (SEM)

*S.* Typhimurium SL1344 was grown on the glass slides at 37 °C for 24 h in the presence of 1 mg/mL of propionate or butyrate. The untreated *S.* Typhimurium SL1344 was used as a control. After incubation for 48 h, planktonic bacteria were removed, and biofilm cells were washed with 0.01 M PBS, and subsequently fixed with 2.5% (*v*/*v*) glutaraldehyde overnight at 4 °C. The biofilms were then washed three times with 0.01 M PBS, and dehydrated with a series of ethanol (10%, 30%, 50%, 70%, 80%, 90%, and 100% *v*/*v*). After drying at 60 °C for 4 h, the glass was sputter-coated with gold under vacuum conditions. The biofilm was visualized using a scanning electron microscope (Hitachi, Tokyo, Japan).

### 2.7. XTT Reduction Assay

Biofilm metabolic activity was measured by XTT (2,3-Bis(2-methoxy-4-nitro-5-sulfophenyl)-2-Htetrazolium-5-carboxamide, Aladdin, Shanghai, China) reduction assay as described previously [5]. Briefly, *S.* Typhimurium SL1344 was incubated in the presence of 1 mg/mL of propionate or butyrate. The untreated *S.* Typhimurium SL1344 was used as a control. After incubation at 37 °C for 24 h or 48 h, the unbound cells were washed with 0.01 M PBS twice, and 50 μL of activated XTT solution was added to each well. After incubation at 37 °C for 2 h, the developed color was measured at OD492 nm by a microplate reader.

### 2.8. Fluorescence Microscopic Analysis

*S.* Typhimurium SL1344 was grown on the glass slides at 37 °C for 24 h in the presence of 1 mg/mL of propionate or butyrate. The untreated *S.* Typhimurium SL1344 was used as a control. After incubation for 24 h, planktonic bacteria were removed, and biofilm cells were washed twice with 0.01 M PBS. The formed biofilm was stained using SYTO 9/PI live/dead bacterial double stain kit (Maokang, Shanghai, China). After incubation for 25 min, the unbound colorant was rinsed with 0.01 M PBS twice. The images were visualized by fluorescence microscope (Nikon, Tokyo, Japan) at ×10 magnification. Image J calculated the fluorescence intensities of live and dead cells. The results were represented as % of dead cells.

### 2.9. Quantitative QS Inhibition Assay

A bioluminescent bacterial reporter strain called *V. harveyi* BB170 was used in the assay, which produces light in response to AI-2 produced by *S.* Typhimurium SL1344 [24]. First, the effect of propionate or butyrate at SICs on the growth of *V. harveyi* BB170 was determined as described in Section 2.4. Then, the SIC of propionate or butyrate against *V. harveyi* BB170 was selected for the QS inhibition assay.

The *V. harveyi* BB170 was used to determine the effect of two kinds of SCFAs on AI-2 production [25]. *S.* Typhimurium SL1344 was prepared as described in Section 2.2. Bacteria were incubated with LB broth supplemented with various concentrations of propionate or butyrate (0.125 mg/mL and 0.25 mg/mL) and incubated at 37 °C for 6 h. The untreated *S.* Typhimurium SL1344 was used as a control. After incubation, the cultures were centrifuged at 5000× *g* for 5 min at 4 °C to obtain the supernatant containing QS molecules. The supernatant was passed through 0.22 μm filters and stored at −20 °C for use. *V. harveyi* BB170 was cultured overnight in autoinducer bioassay (AB) broth and was diluted to OD600 nm = 0.2, then 4 mL of *V. harveyi* BB170 was mixed with 1 mL of supernatant. The mixture was incubated at 37 °C for 24 h with shaking at 120 rpm. Then, luminescence was measured using a microplate reader (Tecan, Männedorf, Switzerland).

### 2.10. Swimming Motility Assay

Motility activity was evaluated in LB broth containing 0.3% agar concentrations as previously described [18]. Both SCFAs were added to LB broth at a final concentration of 0 mg/mL, 0.25 mg/mL, 0.5 mg/mL, 1 mg/mL, and 2 mg/mL, then the plates were placed at room temperature for 40 min. The semi-solid agar plates were spotted with 2 μL volumes of *S.* Typhimurium SL1344 (10^6^ CFU/mL) at the center of the plates and incubated at 37 °C for 12 h or 24 h, respectively. Medium without SCFAs was used as a control.

### 2.11. Adhesion to and Invasion of Caco-2 Cells

The effects of propionate or butyrate on the adhesion and invasion of *S.* Typhimurium SL1344 were carried out according to the method described previously [26]. The colorectal adenocarcinoma (Caco-2) cell line was obtained from the Cell Bank of the Chinese Academy of Sciences (TCHu146, Shanghai, China) and grown in Dulbecco’s Modified Eagle Medium (DMEM) (Gibco, New York, NY, USA) supplemented with 20% (*v*/*v*) fetal bovine serum (Hyclone, South Logan, UT, USA) and 1% (*v*/*v*) double antibiotic solution (100 U/mL of penicillin and 100 μg/mL of streptomycin, Hyclone, South Logan, UT, USA) at 37 °C and 5% CO_2_ for 18 h. Then Caco-2 was seeded in a 48-well plate at a final concentration of 10^4^ cells/well. For the treatment of *S.* Typhimurium SL1344, the strain was cultured with propionate (0.5 mg/mL and 2.0 mg/mL) or butyrate (0.5 mg/mL and 2.0 mg/mL) for 8 h, then centrifuged at 8000× *g* for 5 min to remove residual propionate or butyrate and re-suspended in DMEM at a final concentration of 10^6^ CFU/mL. The untreated *S.* Typhimurium SL1344 was used as a control. After that, treated *S.* Typhimurium SL1344 was co-incubated at a multiplicity of infection (MOI) of 10 with cells at 37 °C for 1 h in a humidified 5% CO_2_ incubator.

For the adhesion assay, the infected monolayers of Caco-2 were washed twice with 0.01 M PBS, added to 1 mL of 0.1% Triton X-100 (Amresco, Solon, OH, USA), and placed at 4 °C for 20 min. The adherent of *S.* Typhimurium SL1344 was plated onto LB agar. For the invasion assay, the infected Caco-2 monolayers were washed twice with 0.01 M PBS, then incubated at 37 °C for 1 h in 1 mL of DMEM containing gentamicin (100 μg/mL). Residual gentamicin was washed twice, then each well was lysed with 0.1% Triton X-100 at 4 °C for 20 min. The invasive *S.* Typhimurium SL1344 was plated in LB plate. The adherent and invasive rates in the control group *S.* Typhimurium were set at 100%, and the rates in treated groups were calculated as a percentage of the control.

### 2.12. Quantitative Real-Time PCR

*S.* Typhimurium SL1344 was treated with propionate or butyrate (0.5 mg/mL and 2.0 mg/mL) for 8 h, and the bacteria were centrifuged at 8000× *g* for 5 min, then bacteria were harvested for use. The untreated *S.* Typhimurium SL1344 was used as a control. The total RNA of bacteria was extracted using TRIzol reagent, then the total RNA was reverse-transcribed to cDNA using PrimeScript™ RT reagent Kit (Takara, Dalian, China) following the manufacturer’s instructions. The expression of target genes was obtained by TB Green^®^ Premix Ex Taq™ II (Takara, Dalian, China), and relative gene expression was quantified by a Real-Time PCR system (Applied Biosystems, Carlsbad, CA, USA). The 16S rRNA gene was used as the internal reference gene. The relative gene expression was analyzed by the 2^−ΔΔCt^ method. Primers’ information was listed in Table 1.

### 2.13. Statistical Analysis

All data presented in this study are expressed as mean ± standard deviation. Statistical analyses were performed by one-way analysis of variance (ANOVA) by SPSS 23.0 (SPSS, Los Angeles, CA, USA). “*”, “**” and “***” indicates significance compared to the control. “^#^”, “^##^” and “^###^” indicates significance between different SCFAs treatments at the same concentration SCFAs. “*” “^#^” *p* < 0.05, “**” “^##^” *p* < 0.01, “***” “^###^” *p* < 0.001. All figures were drawn by Origin 2019b (Origin, Hampton, VA, USA).

## 3. Results

### 3.1. MICs and SICs

The MICs for propionate and butyrate against *S.* Typhimurium were 64 mg/mL and 32 mg/mL, respectively. According to the growth curves, when the concentrations of SCFA were below 4.0 mg/mL, there is no significant difference compared to the control shown in Figure 1. Therefore, concentrations below 4.0 mg/mL (2, 1, 0.5, 0.25, and 0.125 mg/mL) were chosen as SICs in further experiments.

### 3.2. Biofilm Reduction

Firstly, we investigated the biofilm formation of *S.* Typhimurium SL1344 in LB broth. When *S.* Typhimurium SL1344 was cultured with propionate or butyrate at 37 °C in LB, biofilm formation was considerably suppressed (*p* < 0.001) after 24 h, showing a dose-dependent manner (Figure 2A). After 48 h, a significant reduction was observed for two SCFAs treated bacteria (Figure 2B). Moreover, SEM analysis also demonstrated visually the inhibitory effect of SCFAs on the microstructure of biofilm. Biofilm was comprised of a dense layer in the absence of SCFAs, whereas the biofilms were much less dense when SCFAs were added into the broth (Figure 2C). Those results suggest that SCFAs effectively interfere with the biofilm formation of *S.* Typhimurium SL1344 in LB broth.

Then, two SCFAs were added to the food models (milk or chicken broth) to explore inhibitory effects on biofilm. As shown in Figure 3A,B, biofilm formation of *Salmonella* was significantly reduced by 2 mg/mL of propionate or butyrate to 72.16% and 57.56% of control (*p* < 0.01), respectively, after 24 h in the milk model, while we only observed biofilm reduction in butyrate-treated bacteria after 48 h incubation. Biofilm formation was also carried out in chicken broth. After incubation at 37 °C for 24 h, the formed biofilm was 88.51% and 71.47% of the control, respectively, in the exposure to 1 mg/mL of propionate or butyrate. The formed biofilm was 72.09% and 55.46% of the control, respectively, when bacteria were treated with 2 mg/mL of propionate or butyrate (Figure 3C). After incubation at 37 °C for 48 h, no biofilm inhibition was observed for propionate at all tested concentrations, while biofilm reduction was still seen at 2 mg/mL of butyrate-treated groups (*p* < 0.001) (Figure 3D).

### 3.3. Biofilm Metabolic Activity

The suppression of propionate or butyrate against the biofilm metabolic activity of *S.* Typhimurium SL1344 was also investigated. After 24 h incubation, the metabolic activity of the biofilm had a substantial reduction after treatment with propionate or butyrate, and the inhibitory effect of butyrate was significantly stronger than that of propionate in Figure 4A (*p* < 0.001). However, the inhibitory effect of both SCFAs decreased with time, and no inhibitory effect was observed after 48 h (Figure 4A). Furthermore, the fluorescence staining verified that both propionate and butyrate markedly increased the ratio of the dead cell (red) inside the biofilm after incubation at 37 °C for 24 h (Figure 4B,C).

### 3.4. AI-2 Quorum Sensing

The inhibition of quorum sensing was assessed indirectly using a reporter strain *V. harveyi* BB170. First, we confirmed the SICs of propionate or butyrate. There is no apparent inhibitory effect on the growth of *V. harveyi* BB170 at 0.25 mg/mL and 0.125 mg/mL for propionate or butyrate (Figure 5A,B). The production of AI-2 of *S.* Typhimurium SL1344 was reduced by 12.97% and 29.11% when exposed to propionate at 0.125 mg/mL and 0.25 mg/mL, and by 20.55% and 32.87% when exposed to butyrate at 0.125 and 0.25 mg/mL, respectively (*p* < 0.05) shown in Figure 5C. The inhibitory effect of butyrate on AI-2 production was significantly higher than that of propionate at a concentration of 0.125 mg/mL (*p* < 0.05), while no difference was observed at a concentration of 0.25 mg/mL.

### 3.5. Swimming Motility

As shown in Figure 6A,B, the swimming motility of *S.* Typhimurium SL1344 was reduced by butyrate. The swimming area of untreated *S.* Typhimurium SL1344 was 8.99 ± 1.06 cm^2^ after 12 h, whereas that of bacteria treated with 0.5 mg/mL, 1.0 mg/mL, and 2.0 mg/mL of butyrate were 6.76 ± 1.01 cm^2^, 6.08 ± 0.51 cm^2^, and 5.26 ± 0.41 cm^2^, respectively (about 75.18%, 67.62% and 58.44% of the control, respectively). The same trend was observed at 24 h. However, no significant inhibitory effect on swimming motility was observed when *S.* Typhimurium SL1344 was exposed to propionate.

### 3.6. Bacterial Adhesion and Invasion

Propionate or butyrate significantly inhibited the ability of *S.* Typhimurium to adhere to and invade Caco-2 cells in a dose-dependent manner (*p* < 0.01) shown in Figure 7. Meanwhile, we found that propionate-treated bacteria had lower adhesion ability than butyrate-treated bacteria at 1 mg/mL and 2 mg/mL. While the inhibitory effects on invasion were stronger for butyrate in contrast to propionate (*p* < 0.05) at all tested concentrations.

### 3.7. Genes Expression

The expression of biofilm formation-related genes (arcZ, adrA, csgD, and pipB) and invasion-related genes (sipA, sipB, sipc, and hilD) were examined by RT-PCR, as shown in Figure 8. The expressions of arcZ and adrA were reduced significantly after treatment with propionate or butyrate at both two tested concentrations, while the expression of csgD and pipB increased after the incubation with 2 mg/mL of propionate or butyrate. A significant reduction of gene expression (*p* < 0.05) was observed for sipA, sipB, sipC, and hilD for both two SCFAs, especially for butyrate at 2 mg/mL.

## 4. Discussion

*Salmonella* commonly contaminates various types of food and causes a large number of infections globally [27]. Various virulence traits such as biofilm formation, motility, and invasion contribute to bacterial persistence in the environment and the infection process. SCFAs are widely found in the mammalian gut and contribute to intestinal homeostasis in vivo [28,29,30]. However, most studies have focused on in vivo health effects of propionate or butyrate, and their impact on foodborne pathogens and potential application in the food industry has not been extensively explored. In this study, we examined the effect of two common SCFAs on the biofilm formation of *Salmonella* in different media and on other virulence traits. Moreover, we also determined the expression of genes related to biofilm and invasion.

*Salmonella* could form biofilms on various surfaces in the food industry, including foods, stainless steel, aluminum, plastic, and glass [31]. Furthermore, the biofilm lifestyle of *Salmonella* is more advantageous for persistence in the host and the environment compared to planktonic cells [8]. Multiple researches verify that the antibacterial effect of lactic acid bacteria stems from organic acid, especially lactic acid. Organic acid such as lactic acid produced by *Lactobacillus* has been applied in food processing as a natural preservative [5]. Moreover, the capacity of other organic acids to inhibit bacterial growth and virulence is well established [32]. Agaric acid has been reported to dramatically reduced *Salmonella* biofilm formation [33]. The chlorogenic acid combined with ultrasound showed strong antibacterial and antibiofilm effects on the formed biofilm of *Salmonella* and significantly decreased polysaccharides of biofilm [34]. Ferulic acid and p-coumaric acid decreased biofilm formation without growth inhibition and repressed the biofilm formation-related gene expressions of *S. Enteritidis* [35]. Except for *Salmonella*, the antibacterial and antibiofilm effect of various organic acids were also observed in other pathogens, including ferulic and gallic acids against *L. monocytogenes* and *E. coli* [36], chlorogenic acid against *Y. enterocolitica* [37], chlorogenic acid against *S. aureus* [38]. As organic acids, both propionate and butyrate exerted a significant inhibitory effect on the biofilm formation in LB broth in our study (Figure 2). It has been shown that the biofilm formed under laboratory media may differ from that formed in the food model [16,19,39]. Compared to previous studies examining the biofilm formation of *S.* Typhimurium in laboratory conditions, we also proved the inhibitory effect of two SCFAs in the milk model and chick broth model. However, the anti-biofilm effect was obvious in LB broth for two SCFAs during the whole incubation, while this inhibitory effect existed after 24 h incubation and was reduced at 48 h in the food model. We hypothesized that the variation of nutritional composition and ingredients in food might be contributed to the reduction of inhibition at 48 h, while butyrate at 2 mg/mL still exhibited an inhibitory effect even after 48 h, indicating its better potential than propionate for future application (Figure 3).

Motility is fundamental for *Salmonella* to adhere to a surface [12]. The flagellum plays a vital role in its motility, and studies have suggested that flagella may act as motion providers and surface binders during biofilm formation [40,41]. The reduced motility and slowed flagella motion are connected with the disruption of intracellular pH homeostasis [17,42]. Phenolic acids and gallic acid exhibited total inhibition of swimming motility and swarming against *L. monocytogenes* [36]. In addition, it is also possible that the inhibited motility is due to the effect of butyrate on gene transcription in *Salmonella*. For example, gene arcZ regulates motility and/or chemotaxis of *Salmonella*, which was closely associated with the surface attachment of bacteria [43]. The expression of arcZ in *S.* Typhimurium SL1344 was significantly inhibited by butyrate and propionate. Similarly, the motility of *Salmonella* was significantly inhibited, and the transcription of flagellar genes (arcZ) was restricted in the milk model [18]. The swimming ability was significantly decreased when exposed to SICs ferulic acid and p-coumaric acid, and it was proved that related gene expression of *S. Enteritidis* was repressed [35]. Moreover, motility is an important role in *Salmonella* adhering to biotic or unbiotic surfaces [44]. On the other side, we observed the adhesion and invasion gene of *Salmonella* decreased, which is consistent with the result of motility.

Quorum sensing (QS) is a cell-to-cell communication system depending on small signaling molecules which are called autoinducers (AIs) [45]. The anti-virulence strategy suggested includes the disturbance of QS through different methods, such as the intervention of AI-2 synthesis and inhibition of protein in the QS system [46]. Moreover, the recent study focus on the strategy of combining QS inhibitors with antibiotics, which is a promising strategy for inhibiting the production of AI-2 and QS activity, while antibiotics have increased the resistance of pathogens [47]. *Salmonella* is capable of producing AI-2, and it has been reported that AI-2 is involved in the biofilm production, motility, and virulence of *Salmonella* [48]. In our study, AI-2 production might be interfered with after the treatment of propionate or butyrate, while the mechanism might be explored in-depth study. In concordance with the QS results, we also observed that the biofilm decreased in the presence of two kinds of SCFAS.

ArcZ, adrA, and csgD are genes involved in *Salmonella* biofilm formation and have been recognized as key regulators for biofilm formation [49]. ArcZ can regulate biofilm phenotypes in *Salmonella* [43]. It was reported that gene csgD is partially regulated by arcZ, and confirmed that the synthesis of csgD is associated with biofilm formation [43]. We found that the expression of arcZ decreased in the presence of two SCFAs for two concentrations (0.5 mg/mL and 2 mg/mL), while exposure to high concentrations of SCFAs up-regulated the expression of csgD, which needs further exploration. Another study indicated that adrA is a gene encoded by a di-guanylate cyclase and could produce a second messenger c-diGMP, which is an important molecule in regulating biofilm and motility [50]. Similar to the relation between inhibited biofilm and the reduced expression of adrA in this study, previous research also reported that biofilm reduction is associated with the down-regulation of adrA [51]. Meanwhile, it has been proved that the genes (ArcZ, adrA) played important roles in *Salmonella* biofilm in meat thawing loss broth [9].

## 5. Conclusions

In this study, we demonstrated that either propionate or butyrate at SICs could inhibit the biofilm formation of *S.* Typhimurium grown in laboratory media and food broths. Moreover, butyrate, better than propionate, could reduce quorum sensing, bacterial motility, and invasion ability. These findings indicate the potential application of SCFAs such as butyrate in the mitigation of *Salmonella* biofilm in food systems. However, the tested temperature (37 °C) might not be commonly encountered in an applied environment throughout the food chain, and other limitations stemmed from the fact that the concentrations used in practice may differ from those in laboratory conditions. Therefore, we will also focus on the applied temperatures of SCFAs to verify the inhibitory profiles of both SCFAs and the effect of the used concentration on the sensory and other properties of the food.

## Figures and Tables

**Figure 1 foods-11-03493-f001:**
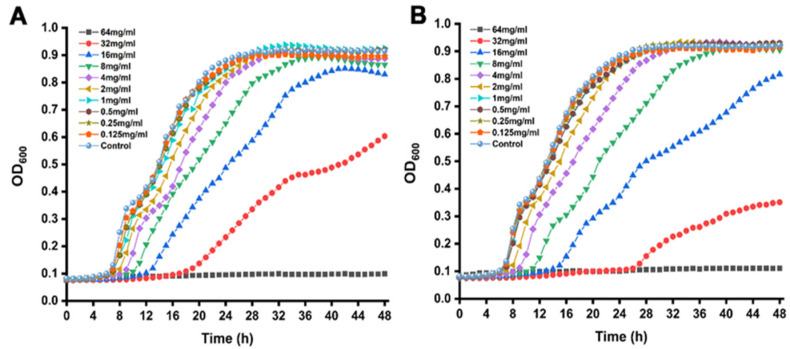
Growth curve of *S.* Typhimurium SL1344 in the presence of (**A**) propionate and (**B**) butyrate.

**Figure 2 foods-11-03493-f002:**
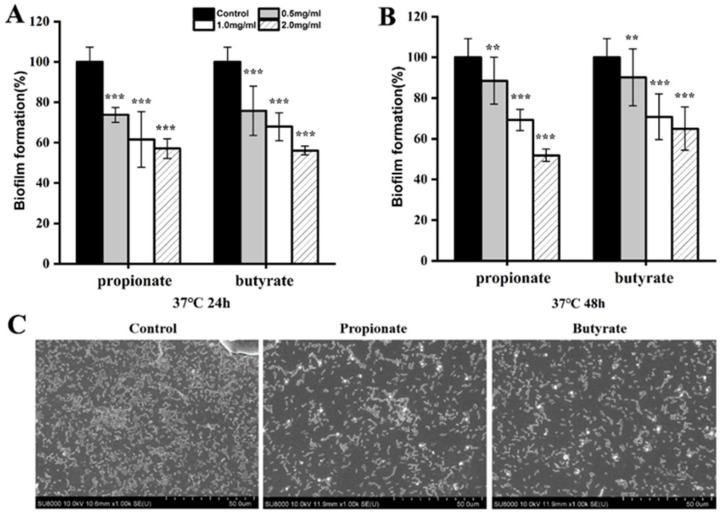
Biofilm formation of *S.* Typhimurium SL1344 grown in LB for (**A**) 24 h at 37 °C and (**B**) 48 h at 37 °C. (**C**) The SEM of biofilm at 1000× magnification. “**” and “***” indicates significance compared to the control. “**” *p* < 0.01, “***” *p* < 0.001.

**Figure 3 foods-11-03493-f003:**
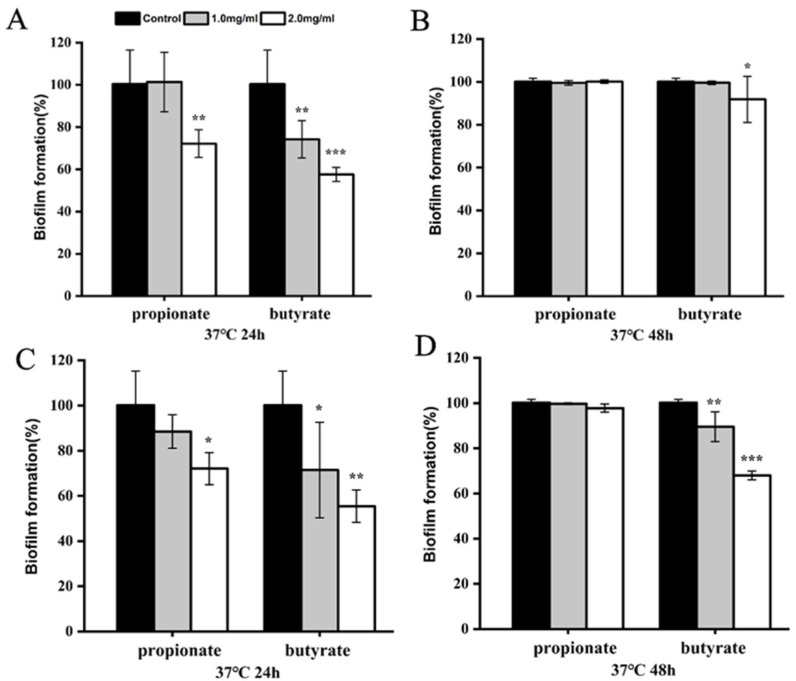
Biofilm formation of *S.* Typhimurium SL1344 grown in milk model at 37 °C for (**A**) 24 h and (**B**) 48 h, or in chicken broth for (**C**) 24 h, (**D**) 48 h. “*”, “***”, and “***” indicates significance compared to the control. “*” *p* < 0.05, “**” *p* < 0.01, “***” *p* < 0.001.

**Figure 4 foods-11-03493-f004:**
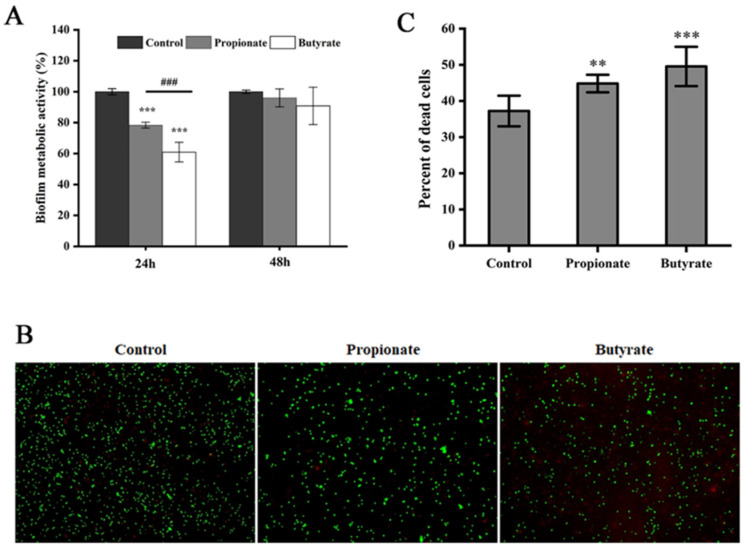
The biofilm metabolic activity of *S.* Typhimurium SL1344 was analyzed by (**A**) XTT assay, (**B**) fluorescence staining, and (**C**) quantitative analysis by Image J. “**”, and “***” indicates significance compared to the control. “^###^” indicates significance between different SCFAs treatments. “**” *p* < 0.01, “***” “^###^” *p* < 0.001.

**Figure 5 foods-11-03493-f005:**
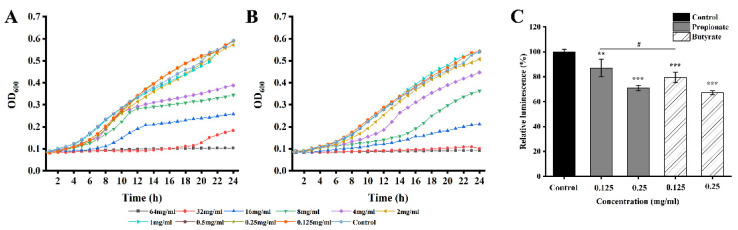
The growth curve of *V. harveyi* BB170 treated with different concentrations of (**A**) propionate and (**B**) butyrate, respectively. (**C**) The AI-2 production of *S.* Typhimurium SL1344. “**” and “***” indicates significance compared to the control. “^#^” indicates significance between different SCFAs treatments. “^#^” *p* < 0.05, “**” *p* < 0.01, “***” *p* < 0.001.

**Figure 6 foods-11-03493-f006:**
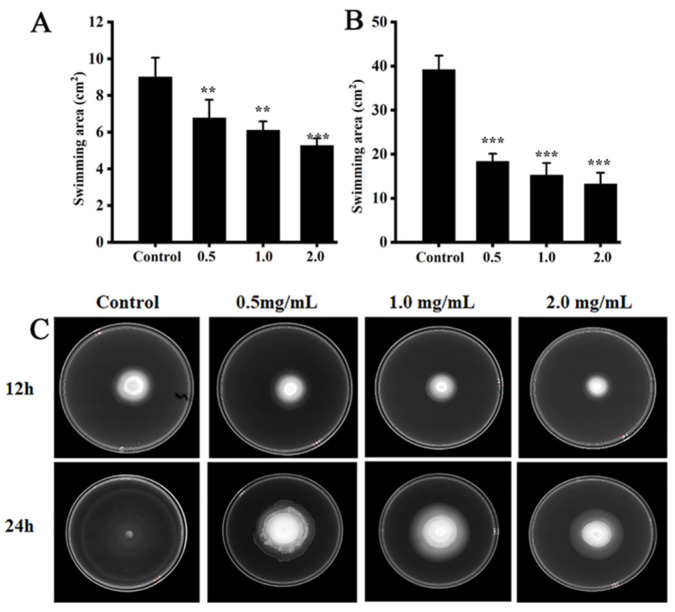
Effects of butyrate on the swimming motility of *S.* Typhimurium SL1344 at (**A**) 12 h, (**B**) 24 h, and (**C**) representative images of swimming motility. “**” and “***” indicates significance compared to the control. “**” *p* < 0.01, “***” *p* < 0.001.

**Figure 7 foods-11-03493-f007:**
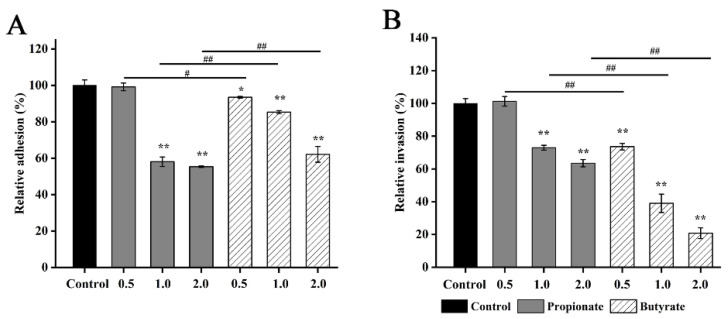
Effect of SCFAs on the (**A**) adhesion to and (**B**) invasion of *S.* Typhimurium SL1344 to Caco-2 cells. “*” and “**” indicates significance compared to the control. “^#^” and “^##^” indicates significance between different SCFAs treatments. “*” “^#^” *p* < 0.05, “**” “^##^” *p* < 0.01.

**Figure 8 foods-11-03493-f008:**
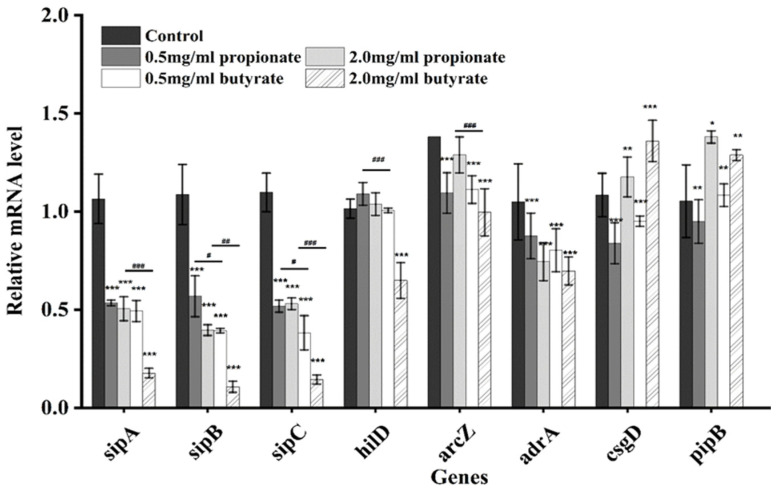
Relative expression of genes related to biofilm formation and invasion. “*”, “**”, and “***” indicates significance compared to the control. “^#^”, “^##^” and “^##^” indicates significance between different SCFAs treatments. “*” “^#^” *p* < 0.05, “**” “^##^” *p* < 0.01, “***” “^###^” *p* < 0.001.

**Table 1 foods-11-03493-t001:** Primers used in RT-PCR.

NO.	Gene	Primers
1	SipA	CGCTGTCAGGGGAAATTAAA
		ATTATCGCTTTCTTACCGGC
2	SipB	GCCGATGAAATTGTGAAGGC
		CCTAATCCTTCCAGCGCTTT
3	SipC	GAATAAATCCCGCCGCTTAT
		GGTCACTGACTTTACTGCTG
4	arcZ	ACTGCGCCTTTGACATCATC
		CGAATACTGCGCCAACACCA
5	csgD	TCCTGGTCTTCAGTAGCGTAA
		TATGATGGAAGCGGATAAGAA
6	adrA	GAAGCTCGTCGCTGGAAGTC
		TTCCGCTTAATTTAATGGCCG
7	pipB	GCTCCTGTTAATGATTTCGCTAAAG
		GCTCAGACTTAACTGACACCAAACTAA
8	HilD	TAACGTGACGCTTGAAGAGG
		GGTACCGCCATTTTGGTTTG
9	16s rRNA	AGGCCTTCGGGTTGTAAAGT
		GTTAGCCGGTGCTTCTTCTG

## Data Availability

Data are contained within the article.

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
