# Peer review of "Propionate and Butyrate Inhibit Biofilm Formation of Salmonella Typhimurium Grown in Laboratory Media and Food Models"

_foods, 2022, doi:10.3390/foods11213493_

Round 1

Reviewer 1 Report

The manuscript "Propionate and butyrate inhibit biofilm formation of Salmonella Typhimurium grown in laboratory media and food models" presents interesting research results, but requires very large corrections.

Detailed comments:

The manuscript is not prepared according to the template and guidelines of the Foods journal. It has an incorrect layout, incorrect way of citing literature. Figures and Tables should be in the main text, not at the end.

Abstrct requires editing and adding research results.

Keywords are chosen correctly.

Lines 36-42 - highlights are not in MDPI journals.

In the introduction, the media simulating the food system should be described in more detail.

Chapter 2.2 - please describe in more detail the origin of bacterial strains. What public collection do they come from? If they are isolates, how have they been identified?

Chapter 2.3 - fresh milk was used - this is not a substrate simulating a food product, but the actual food product.

Chapter 2.4 - On what basis was the 630 nm wavelength used? This is not a standard wavelength for bacteria testing. Bioscreen C uses 100-well plates, not 96-well. Please describe in more detail the performance of the MIC determination (whether it is a micro or macro dilution method), this test is performed at a specific time (for bacteria it is 24h) and at a specific inoculum concentration. Please refer to the methodology used for the analysis (EUCAST or NCCLS).

The results are well described, the discussion is correct.

The References chapter is not prepared according to the guidelines of the Foods. There is little recent literature (from the last 2-3 years), authors should consider the latest publications in discussing the results.

Figure 1 - the methodology describes that the experiment lasted 48 hours, while the figure shows 24 hours.

Figure 5, QS determination - why was this experiment performed and a different bacterium introduced than in the other experiments?

Reviewer 2 Report

Journal: foods

Title: Propionate and butyrate inhibit biofilm formation of Salmonella  Typhimurium grown in

          laboratory media and food models

Authors: Liu et al., 2022.

Manuscript No.: foods 1912708

The manuscript titled ‘Propionate and butyrate inhibit biofilm formation of Salmonella Typhimurium grown in laboratory media and food models’ was reviewed. The central idea of the research is well hypothesized and the research work was carried out meticulously in a systematic manner with scientific accuracy. The manuscript is well compiled with a brief Introduction quoting recent research and review articles. The materials and methods are precise and comprehensive. The results and discussion part is satisfactory. 

            The following parts need a slight clarification

      The binomial Salmonella Typhimurium should be clarified. The species name is represented with capital letter. To exactly imply, The used bacterial culture, Salmonella Typhimurium SL 1344, is Salmonella enteric serovar Typhimurium. A clarity with the representation of the binomial should be given

      In determination of MIC, the minimum concentration of the SAC’s used is 0mg/ml and maximum concentration used is 64mg/ml. Why and how was the maximum concentration fixed at 64mg/ml?

      Quote the reason for selection of propionate and butyrate SCFA’s when there are other SCFA’s available?

      The reference section is not uniform. Arrange the references to the format of the journal.

The authors have put in great effort in formulating and conducting the research and writing and compiling the manuscript which allures sincere appreciation.

Reviewer 3 Report

Consider evaluating more bacteria (same genus and species) but isolated from different sources and reporting effect

How could we apply fatty acids at all points of the possible development of biofilms?

Until what concentration would its use be possible?

Expand the reference section with more recent citations

The discussion section is very short, it should be expanded 

Round 2

Reviewer 1 Report

Manuscript has been revised according to the reviewer's comments. The authors' answers are sufficient. I have no other comments on the current version.